# Interactions between Intestinal Homeostasis and NAD^+^ Biology in Regulating Incretin Production and Postprandial Glucose Metabolism

**DOI:** 10.3390/nu15061494

**Published:** 2023-03-20

**Authors:** Taichi Nagahisa, Shotaro Kosugi, Shintaro Yamaguchi

**Affiliations:** Division of Endocrinology, Metabolism and Nephrology, Department of Internal Medicine, Keio University School of Medicine, Shinjuku-ku, Tokyo 160-8582, Japan

**Keywords:** incretin, nicotinamide adenine dinucleotide, nicotinamide phosphoribosyltransferase, postprandial glucose metabolism, intestinal homeostasis

## Abstract

The intestine has garnered attention as a target organ for developing new therapies for impaired glucose tolerance. The intestine, which produces incretin hormones, is the central regulator of glucose metabolism. Glucagon-like peptide-1 (GLP-1) production, which determines postprandial glucose levels, is regulated by intestinal homeostasis. Nicotinamide phosphoribosyltransferase (NAMPT)-mediated nicotinamide adenine dinucleotide (NAD^+^) biosynthesis in major metabolic organs such as the liver, adipose tissue, and skeletal muscle plays a crucial role in obesity- and aging-associated organ derangements. Furthermore, NAMPT-mediated NAD^+^ biosynthesis in the intestines and its upstream and downstream mediators, adenosine monophosphate-activated protein kinase (AMPK) and NAD^+^-dependent deacetylase sirtuins (SIRTs), respectively, are critical for intestinal homeostasis, including gut microbiota composition and bile acid metabolism, and GLP-1 production. Thus, boosting the intestinal AMPK–NAMPT–NAD^+^–SIRT pathway to improve intestinal homeostasis, GLP-1 production, and postprandial glucose metabolism has gained significant attention as a novel strategy to improve impaired glucose tolerance. Herein, we aimed to review in detail the regulatory mechanisms and importance of intestinal NAMPT-mediated NAD^+^ biosynthesis in regulating intestinal homeostasis and GLP-1 secretion in obesity and aging. Furthermore, dietary and molecular factors regulating intestinal NAMPT-mediated NAD^+^ biosynthesis were critically explored to facilitate the development of new therapeutic strategies for postprandial glucose dysregulation.

## 1. Introduction

Postprandial hyperglycemia in impaired glucose tolerance (IGT) is a crucial risk factor for type two diabetes and cardiovascular diseases (CVDs) [1,2,3,4,5,6]. The Funagata study conducted on a Japanese cohort revealed that IGT is a risk factor for CVDs and stroke [7]. In addition, the Baltimore Longitudinal Study of Aging, a long-term follow-up study of an adult caucasian population, showed that IGT, and not impaired fasting glucose tolerance, elevates the risk of coronary heart disease [8]. However, the Study To Prevent Non-insulin Dependent Diabetes Mellitus (STOP-NIDDM) trial demonstrated that the α-glucosidase inhibitor (αGI), acarbose, which specifically prevents postprandial hyperglycemia, reduces the risk of progression of IGT to type two diabetes [9] and CVDs [10]. Therefore, postprandial glucose levels are an important therapeutic target for preventing the progression of type two diabetes and its macrovascular complications. However, conventional antidiabetic medications targeting postprandial hyperglycemia, such as αGI and glinide, cause adverse effects, including abdominal symptoms and hypoglycemia, respectively [11,12]; therefore, the development of new therapeutic strategies has been sought. 

Postprandial plasma glucose concentrations are regulated by gut incretin hormones, such as glucagon-like peptide-1 (GLP-1) and glucose-dependent insulinotropic polypeptide (GIP), which augment insulin secretion when glucose is administered orally; this process is called glucose-stimulated insulin secretion (GSIS), regulating postprandial glucose metabolism [13,14]. GLP-1 is released from L cells, which are mainly located at the ileum, the distal part of the gastrointestinal tract, and is triggered by neuronal or hormonal stimulation and direct nutritional components, including carbohydrates, amino acids, proteins, and fatty acids [15,16,17,18]. GIP is secreted from enteroendocrine K cells in the upper small intestine [19]. Several human studies have shown that the GLP-1 secretory response is impaired in obese subjects with postprandial hyperglycemia, although conflicting results have also been reported [20,21,22]. In addition, fasting GLP-1 and glucose-stimulated GLP-1 secretion decreased significantly over six years in healthy older subjects aged 71.2 ± 3.8 years [23]. Consequently, the intestine and its derived incretin hormones have become potential targets for novel therapeutics to improve postprandial hyperglycemia in obesity and aging. Dipeptidyl peptidase four (DPP4) inhibitors and GLP-1 analogs, which elevate GLP-1 levels, are already commercially available. Additionally, GLP-1 analogs have been found to have preventive effects on CVDs [24]. However, the mechanisms underlying the reduced GLP-1 secretion in obesity and aging, the two major risk factors of insulin resistance and glucose intolerance, remain unclear but might be associated with the alteration of the gut environment modulated by the intake of dietary components. Emerging evidence has suggested that obesogenic diets, including high-fat components, could alter GLP-1 production by affecting the gut microbiota [25,26,27,28,29]. Furthermore, the number of L cells and postprandial plasma GLP-1 secretion decreases in rats fed an obesogenic high-fat diet (HFD) [30]. These findings indicate that fat-rich obesogenic diets alter intestinal homeostasis, which decreases GLP-1 production. Therefore, maintaining intestinal homeostasis could also be a novel strategy for improving GLP-1 production and thereby postprandial glucose metabolism.

We recently analyzed the impact of fat-rich obesogenic diets on intestinal homeostasis and assessed the downstream mediators of diets that affect GLP-1 secretion. It was demonstrated that HFD impairs the biosynthesis of nicotinamide adenine dinucleotide (NAD^+^), a key regulator of cellular energy metabolism, which is mediated by nicotinamide phosphoribosyltransferase (NAMPT) enzymatic activity in intestines. Intestinal NAD^+^ biology plays an important role in intestinal homeostasis and GLP-1 production in the ileum, consequently influencing postprandial glucose levels; boosting intestinal NAD^+^ biosynthesis by administering a key NAD^+^ intermediate, nicotinamide mononucleotide (NMN), augments GLP-1 production under HFD conditions [31]. However, the mechanisms by which obesogenic fat-rich diets impair intestinal NAMPT-mediated NAD^+^ biosynthesis remain elusive. The aim of this review was to analyze the interplay among intestinal NAD^+^ biology, nutritional components, and intestinal homeostasis, including GLP-1 production, and explore the pathophysiological significance and therapeutic potential of intestinal NAD^+^ biology in systemic glucose metabolism. First, recent findings in NAD^+^ research regarding glucose metabolism were analyzed, and our recent study on the novel roles of intestinal NAD^+^ biology in regulating intestinal homeostasis, GLP-1 production, and postprandial glucose metabolism was highlighted. Second, the possible mechanisms by which components, including dietary nutritional contents, affect intestinal homeostasis and NAD^+^ biosynthesis, are discussed. Finally, the clinical application of intestinal NAD^+^ biology as a novel therapeutic target for postprandial hyperglycemia was explored.

## 2. Impact of NAD^+^ Biology on Systemic Glucose Metabolism in an Organ-Specific Manner

### 2.1. Changes in NAD^+^ Biology with Aging and Obesity in Metabolic Organs 

NAD^+^ is crucial for regulating aging, metabolism, cell growth, and inflammation [32,33,34,35]. Although NAD^+^ is utilized as a cofactor or substrate by numerous enzymes to regulate redox status and cellular energy metabolism [36], sirtuins (SIRTs), which are NAD^+^-dependent deacetylases, play essential roles in cellular biological processes to maintain metabolic homeostasis [35,37,38,39]. In rodents, NAD^+^ levels decrease with aging or fat-rich diet-induced obesity in various organs, including the skeletal muscle [40,41,42,43,44], liver [41,42,44,45,46], adipose tissue [41,42,47], brain [48], pancreas [42], heart [46], kidney [46], and lungs [46]. Similarly, in humans, NAD^+^ levels decrease with aging and intake of unbalanced nutrients, such as fat and protein-rich diets, in organs such as the skin [49], skeletal muscle [50], liver [45], and brain [51,52] and in the plasma [53,54] and monocytes [55]. These findings suggest that a decrease in NAD^+^ levels is involved in the pathogenesis of age- and obesity-related diseases in humans and rodents. Decreases in NAD^+^ contents with age and obesity in metabolic organs may be attributed to changes in the expression or activity of NAD^+^ biosynthetic and consuming enzymes [41]. For example, the enzymatic activity of poly (ADP-ribose) polymerase (PARP)1, a representative NAD^+^ degrading enzyme, increases in various organs such as the liver, kidney, skeletal muscle, lung, and heart owing to the accumulation of DNA damage associated with aging and HFD-induced obesity and correlates with age- and obesity-related changes in NAD^+^ levels [46,56,57,58]. 

A cluster of differentiation 38 (CD38), another major NAD^+^-degrading enzyme, is highly expressed in inflammatory macrophages that infiltrate organs during aging [59,60]. Furthermore, aging and HFD-induced obesity increase CD38 expression in macrophages and the vascular endothelium. Therefore, CD38 expression increases with age and HFD-induced obesity in major organs such as the liver, skeletal muscle, and lungs, decreasing NAD^+^ levels [58,59,60,61,62,63]. NAD^+^ levels are maintained in CD38-deficient mice during aging and obesity stress [41,61]. In contrast to the enhanced NAD^+^ consuming pathway, NAD^+^ biosynthesis is impaired with aging and obesity, decreasing NAD^+^ levels. In mammals, the first step of systemic NAD^+^ biosynthesis depends on the salvage pathway in which nicotinamide (NAM), a water-soluble vitamin B3, is converted to NMN by NAMPT. Next, the NMN is converted to NAD^+^ by a second enzyme, nicotinamide mononucleotide adenylyltransferase [58,62] (Figure 1). The expression of NAMPT, the rate-limiting enzyme in the salvage NAD^+^ biosynthetic pathway, decreases with aging and obesity in various organs, including in white and brown adipose tissue, liver, skeletal muscle, hippocampus, and retina [42,47,63,64,65,66].

### 2.2. Impacts of Decrease in NAD^+^ Levels with Aging and Obesity on Metabolic Organ Function

The pathophysiological importance of NAD^+^ biology in metabolic organs has been investigated using genetically engineered mice models. PARP-1 knockout (KO) mice have increased NAD^+^ contents in their skeletal muscle and brown fat tissues, inducing higher mitochondrial content, increased energy expenditure, and protection against glucose intolerance and insulin resistance [57]. In addition, CD38 KO mice exhibit suppression of age-related decreases in NAD^+^ levels in major organs, preserving mitochondrial function and energy and glucose metabolism [41]. For example, the genetic ablation of CD38 increases NAD^+^ levels in brown adipose tissues, enhancing the thermogenic activity of brown adipocytes [67].

The impacts of decreased NAD^+^ levels on various organ dysfunctions have been investigated using genetically engineered *Nampt*-knockout (NKO) mice models. Homozygous NKO mice exhibit embryonic lethality [68], whereas heterozygous NKO mice exhibit IGT due to impaired insulin secretion [68] and heart failure due to progressive mitochondrial dysfunction induced by pressure overload [69]. 

The roles of NAMPT-mediated NAD^+^ biosynthesis in regulating organ function have also been evaluated using conditional NKO mice. Projection neuron-specific NKO mice develop synaptic dysfunction at neuromuscular junctions in the cerebral cortex and motor neuron degeneration-associated muscle atrophy, resulting in death within an average of 22 days [70]. The cortex- and hippocampus-specific NKO mice exhibit cortical and hippocampal atrophy, abnormal neuronal morphology, and impaired memory and cognitive function [66,71]. Moreover, the hippocampal CA1 region-specific NKO impairs cognitive function [72], and rod and cone cell-specific NKO mice exhibit mitochondrial dysfunction and severe retinal degeneration [73]. Furthermore, skeletal muscle-specific NKO mice develop progressive muscle atrophy [43], hepatocyte-specific NKO mice show hepatic inflammation and fibrosis and reduced liver regeneration [74,75], and proximal tubule-specific NKO mice exhibit glomerular basement membrane thickening and interstitial fibrosis, pathological features of diabetic nephropathy [76]. Vascular smooth muscle-specific NKO mice show a dilated aorta, and angiotensin II administration causes intravascular hemorrhage and aortic dissection [77]. 

It was recently demonstrated that decreased NAD^+^ content in white adipose tissue is associated with severe insulin resistance and IGT due to hypoadiponectinemia at least partly via increased phosphorylation of serine 273 and acetylation of peroxisome proliferator-activated receptor (PPAR) γ in pan adipocyte-specific NKO (ANKO) mice [35,78,79]. Brown adipocyte-specific NKO (BANKO) mice were also established, and their metabolic phenotypes were directly compared with that of ANKO mice. Decreased NAD^+^ levels in white and brown adipose tissues were associated with impaired heat production and energy metabolism in brown adipose tissue [80]. 

In addition to these intracellular NAMPT (iNAMPT), another type of NAMPT exists—extracellular NAMPT (eNAMPT) [68]. The interaction between iNAMPT enzymatic activity and eNAMPT secretion, and the role of plasma eNAMPT, especially in the aging process, was recently identified. The iNAMPT in adipose tissue plays an important role in white and brown adipocyte function by regulating NAD^+^ levels and SIRT1 activity, one of the seven mammalian proteins of the SIRT family. Secretion of eNAMPT from the adipose tissue is determined based on the acetylation status of iNAMPT regulated by SIRT1 [81]. Furthermore, the acetylation status of iNAMPT is regulated by another nuclear SIRT, SIRT6, in cancer cells. SIRT6-mediated regulation of eNAMPT release occurs by modulating the iNAMPT acetylation status in cancer cells, contrasting with the SIRT1-mediated secretion of eNAMPT from the adipose tissues [82], although the precise mechanisms for these different regulation patterns are currently being investigated. 

In ANKO mice, a reduction in plasma eNAMPT levels suppresses hypothalamic NAD^+^ levels and SIRT1 activity, resulting in a defect in physical activity. Contrastingly, adipocyte-specific *Nampt* knockin (ANKI) mice exhibit increased plasma eNAMPT, hypothalamic NAD^+^ levels, and SIRT1 activity, enhancing physical activity [81]. Plasma eNAMPT levels decrease with aging in mice and humans, and a positive correlation exists between plasma eNAMPT levels and life expectancy in mice. Plasma eNAMPT levels in old ANKI mice remained higher than those in the control group, and NAD^+^ levels in the hypothalamus, hippocampus, pancreas, and retina remained higher in female ANKI mice. Old ANKI mice have enhanced expression of SIRT1 target genes in the hypothalamus [83,84], which is important for physical activity and sleep quality and maintains memory and learning ability, retinal photoreceptor neurons, and further insulin secretion capacity with better glucose tolerance. Female ANKI mice maintain their body weight, especially fat mass, even during aging, showing a 13.4% increase in healthy life span compared to the control group [85]. A recent report demonstrated that peripubertal-stressed male mice exhibit increased adiposity, which triggers diminished NAMPT protein levels in adipose tissue and decreased levels of circulating eNAMPT, contributing to lifelong reductions in sociability [86]. These results suggest that maintaining both iNAMPT-mediated NAD^+^ biosynthesis in major metabolic organs and plasma eNAMPT, a key effector of adipose-to-brain signaling, by boosting adipocyte iNAMPT-mediated NAD^+^ biosynthesis could improve organ function and whole-body glucose metabolism and promote a healthy physical and mental status.

## 3. Regulators of Intestinal Homeostasis

### 3.1. Intestinal AMPK and NAD^+^ Biosynthesis 

A study demonstrated that systemic *Nampt*-inducible KO mice lost weight owing to intestinal villus atrophy-associated impaired absorption, which resulted in death within approximately 5–10 days after birth [87]. This suggests that intestinal NAD^+^ biosynthesis plays a critical role in intestinal homeostasis. Although the molecular mechanism(s) underlying the association between intestinal NAD^+^ biology and homeostasis remains enigmatic, AMPK has been demonstrated to upregulate NAMPT, boosting NAD^+^ biosynthesis and SIRT activity [88,89,90]. AMPK, an energy sensor that regulates whole-body energy balance [91], is expressed in several metabolic organs, including the liver, brain, adipose tissues, skeletal muscle, and intestine [92,93]. Calorie restriction activates AMPK in intestinal stem cells and increases NAD^+^ levels by stimulating *Nampt* transcription. Thus, activated intestinal NAMPT-mediated NAD^+^ biosynthesis by AMPK further enhances SIRT1 activity in the intestinal stem cells, consequently increasing their number [94]. Intestinal-specific AMPK upregulation maintains intestinal homeostasis, such as barrier function, and muscle protein homeostasis by inducing autophagy during aging, ultimately extending lifespan [95]. 

Furthermore, pharmacological activation of AMPK enhances intestinal barrier function and epithelial differentiation by promoting the key transcription factor, caudal type homeobox two (CDX2), committing cells to intestinal epithelial lineage [96]. In contrast, high glucose contents attenuate intestinal AMPK activation. AMPK diminished in *Psammomys obesus* with insulin resistance and type 2 diabetes, possibly due to disruptions in insulin signaling at the jejunum [93]. A recent report demonstrated that genetic deletion of intestinal AMPK alters gut microbiota and their metabolites, including antimicrobial peptides, resulting in weight gain and IGT with HFD intake [97]. In addition, intestinal epithelial cell-specific deletion of AMPK results in hyperpermeability in the distal colon with a regular chow diet, accompanied by altered microbial composition [98]. Similarly, AMPK depletion in intestinal epithelial cells impairs intestinal barrier function and integrity and epithelial cell migration [96] (Figure 2). Consistent with the findings that intestinal AMPK, which mediates NAMPT activation, is impaired in glucose intolerance, HFD-induced obesity, and aging downregulates the major NAD^+^-generating enzymes, including NAMPT, and reduces intestinal NAD^+^ levels [31,99,100]. These findings suggest that intestinal AMPK could play a central role in maintaining intestinal homeostasis, including microbial composition, as a possible upstream mediator of intestinal NAMPT-mediated NAD^+^ biosynthesis.

To explore the role of intestinal AMPK in whole-body glucose metabolism, metformin, an AMPK activator, could be the best candidate to analyze. Metformin, an antidiabetic drug, represses hepatic glucose production by inhibiting mitochondrial glycerophosphate dehydrogenase and altering hepatic redox status. Additionally, it activates intestinal AMPK [93,106,107,108]. Thus, its beneficial effect on glucose metabolism also depends on intestinal AMPK, which modulates gut microbial composition [97,109]; this is supported by the observation that loss of intestinal AMPK alters gut microbiota composition [98]. Metformin acts on intestinal AMPK, inhibiting the farnesoid X receptor (FXR), the nuclear receptor for maintaining bile acid homeostasis, consequently increasing the bile acid pool and stimulating GLP-1 secretion [110,111,112]. Additionally, metformin enhances GLP-1 secretion without affecting AMPK activity. GLP-1 secretion is stimulated by the activation of the insulin signaling pathway, followed by the wingless-related integration site (Wnt) signaling pathway in L cells [113,114]. Collectively, boosting intestinal AMPK–NAMPT-–NAD^+^ biosynthesis could benefit systemic glucose metabolism by maintaining intestinal homeostasis, including GLP-1 synthesis and microbial composition (Figure 3).

### 3.2. Intestinal Wnt Signaling

Wnt signaling is another key regulator of intestinal homeostasis. Intestinal Wnt signaling, which is regulated by the gut microbiota [122,123], contributes to intestinal stem cell maintenance [124]. A recent study revealed that intestinal stem cell function and regenerative capacity decrease owing to impaired Wnt signaling during aging; this was evidenced by the blunted organoid formation of crypt obtained from aged mice, which was rescued by restoring canonical Wnt signaling [125]. Similarly, the impairment of intestinal stem cell function during aging was rescued by boosting the NAD^+^ biosynthetic pathway by supplementing nicotinamide riboside (NR), an NAD^+^ intermediate [126]. These findings suggest that impairment of Wnt signaling and intestinal NAD^+^ biosynthesis synergistically or independently disrupt intestinal homeostasis. Furthermore, genetic deletion of the Wnt antagonist, Dickkopf (Dkk) 2, which increases intestinal Wnt activity, stimulates GLP-1 production, thereby improving whole-body glucose tolerance [127]. 

Although plausible factors underlying the relationship between intestinal NAD^+^ biosynthesis and Wnt signaling remain unknown, SIRT1 is a possible contributing molecule. Halloway et al., demonstrated that SIRT1 promotes transient and constitutive Wnt signaling [115]. SIRT1 potentiates β-catenin recruitment into the nucleus, where it binds to the T cell factor/lymphoid enhancer factor [116]. Furthermore, SIRT1 regulates the transcription of downstream target genes of Wnt/β-catenin in osteogenesis [117] and adipogenesis [118]. Overall, intestinal NAD^+^–SIRT1–gut microbiota–Wnt signaling could be crucial for intestinal homeostasis and systemic glucose metabolism by regulating GLP-1 production (Figure 3).

## 4. Pathophysiological Roles of Intestinal NAD^+^ Biosynthesis

The intestine is a central regulator of glucose metabolism [128,129,130]. Intestinal epithelial cell-specific *Sirt1* KO mice exhibit glucose intolerance during oral glucose tolerance tests under calorie-restriction conditions [94]. As NAD^+^ is essential for SIRT activity, intestinal NAD^+^ biosynthetic status was investigated. HFD administration impaired intestinal NAMPT-mediated NAD^+^ biosynthesis [31]. To extensively evaluate the roles of intestinal NAMPT-mediated NAD^+^ biosynthesis in intestinal homeostasis and glucose metabolism, an intestinal epithelial cell-specific NKO mouse model (INKO) was established [31]. INKO mice fed a regular chow diet had shorter intestines with fibrotic changes, suggesting that intestinal NAMPT-mediated NAD^+^ biosynthesis helps maintain intestinal homeostasis. Furthermore, an increased glucose excursion with reduced GLP-1 and insulin secretion was observed in INKO mice after oral glucose loading. We investigated the underlying mechanism of reduced GLP-1 in INKO mice. Silencing transcription factor 7-like 2 (TCF7L2), a crucial transcriptional factor in the canonical Wnt signaling, impacted pancreatic β cell mass and impaired GSIS by disrupting vesicle fusion of secretory granules [131,132]. Humans with genetic variants of TCF7L2 are predisposed to diabetes due to changes in GLP-1 and pulsatile insulin secretion [133,134,135]. In addition, functional ablation of TCF7L2 in proglucagon-expressing cells decreases proglucagon expression and GLP-1-positive cells in the gut, reducing plasma GLP-1 secretion [136]. These findings led us to investigate Wnt signaling as a possible downstream mediator connecting intestinal NAD^+^ biosynthesis and GLP-1 production [136,137]. In vitro experiments using STC-1 cells as enteroendocrine cells demonstrated that ICG-001, a selective inhibitor of Wnt/βcatenin signaling, dose-dependently reduced proglucagon gene expression. Furthermore, the concurrent addition of ICG-001 negated the amelioration of GLP-1 secretion by recovering NAD^+^ levels, suggesting that Wnt signaling regulates GLP-1 secretion from L cells as a downstream mediator of NAD^+^ biosynthesis. 

As postprandial hyperglycemia was observed in intestinal epithelial cell-specific *Sirt1* KO mice, SIRTs could be the molecule connecting intestinal NAMPT-mediated NAD^+^ biosynthesis and canonical Wnt signaling to stimulate GLP-1 secretion [31,94] (Figure 2 and Figure 3). PPARβ/δ might also be responsible for the reduced GLP-1 production due to NAD^+^ depletion, as it has been demonstrated to transcriptionally regulate proglucagon expression in enteroendocrine L cells by stimulating the β-catenin/TCF7L2 pathway [138]. Collectively, intestinal NAMPT-mediated NAD^+^ biosynthesis contributes to Wnt signaling, maintaining intestinal homeostasis, including GLP-1 production and postprandial glucose homeostasis (Figure 3).

## 5. Potential Effects of Dietary Habits and Its Associated Gut Environment on Intestinal Homeostasis and GLP-1 Secretion

Several types of food have been demonstrated to modulate intestinal homeostasis. For example, arachnoid acid, found in animal meat, promotes intestinal epithelial regeneration in cases of irradiation injury by activating Wnt signaling [139,140], and dietary vitamin D, found in sun-dried mushrooms and oily fish, such as salmon, facilitates intestinal epithelial cell turnover during bowel resection in rodents [139,140]. A high-fat ketogenic diet, which elevates circulating ketone body levels [141], boosts intestinal stem cell numbers and function, enhancing intestinal stemness [142]. Similarly, obesogenic HFD augments the number and function of intestinal stem cells and enhances their capacity to initiate tumors [143]. In addition, HFD induces dysbiosis, reducing lactobacillus, which produces lactate. Lactate acts on the lactate receptor, G-protein-coupled receptor 81 (GPR81) and augments the proliferative potential of intestinal stem cells [119,144]. Furthermore, HFD in catch up growth rat models induces lipotoxicity in the intestinal L cells, increasing intestinal L cell apoptosis and reducing GLP-1 secretion [30].

Dietary components are associated with postprandial GLP-1 secretion in humans as well. For example, habitual added sugar consumption exerts a positive effect on striatal response to food cues and a negative effect on postprandial GLP-1 response. These sugar consumption-associated alterations could contribute to pathological overeating [145]. Oral intake of whey protein dose-dependently increases GLP-1 release in younger healthy lean males and healthy older people [146,147]. Relatively large amounts of fat (40% kcal fat diet) increase postprandial GLP-1 secretion in healthy elderlies though not in younger people [148]. Fat ingestion before a carbohydrate meal slows gastric emptying, delays the postprandial rises in blood glucose, plasma insulin, and GIP, and stimulates GLP-1 secretion in type 2 diabetes [149]. These findings support the previous notion that the effect of fat on gastric emptying and absorption of nutrients depends on when it is consumed [150]. The effect of the intake sequence of food macronutrients, including vegetables, protein, and carbohydrates, on postprandial glucose levels, insulin, and incretin secretions was evaluated in healthy adults. The postprandial glucose and insulin response was attenuated, whereas GLP-1 secretion was stimulated by the vegetable-protein-carbohydrate food intake sequence [151].

Another important factor that regulates the interplay between food, intestinal homeostasis, and GLP-1 secretion is the gut microbiota and its associated bile acid abundance and composition. The important role of microbiota in maintaining intestinal homeostasis and regulating whole-body glucose and energy metabolism has been intensely investigated [152,153,154,155]. The microbiota plays a critical role in intestinal immunity, and dysbiosis may be involved in the pathogenesis of obesity and metabolic syndrome by inducing low-grade inflammation in remote organs such as the liver or adipose tissue [156]. However, an alteration of the intestinal immune response leads to obesogenic microbiota and obesity [157]. HFD intake increases the proportion of lipopolysaccharide-containing gut microbiota, triggering metabolic endotoxemia, a low-grade inflammation factor [158]. The underlying mechanisms of such HFD-induced systemic metabolic endotoxemia due to dysbiosis might involve low-grade intestinal inflammation and enhanced gut permeability [159]. Intestinal NAD^+^ biosynthesis and NAD^+^-dependent deacetylase SIRTs, specifically SIRT1, is a possible molecular mechanism underlying the association between age- and obesity-associated alteration of microbiota composition, intestinal inflammation, permeability, and metabolic complications. A previous report also demonstrated that treatment with resveratrol, a potent SIRT1 activator metabolized by gut microbiota, alters microbiota composition, decreases body weight, and improves insulin sensitivity and lipid metabolism in rodents [160,161]. Contrarily, genetic ablation of *Sirt1* in the intestinal epithelium deteriorates age-associated intestinal inflammation and dysbiosis partly by modulating bile acid metabolism [101]. Our previous study also showed that INKO mice have a different gut microbiota composition than their counterparts under a regular chow diet, suggesting that intestinal NAD^+^ biosynthesis is involved in maintaining gut microbiota homeostasis and possibly contributing to the reduced GLP-1 production (unpublished data). Contrastingly, normal gut microbiota variation produces NMN, enhancing pancreatic NAD^+^ biosynthesis and mitochondrial deacetylase SIRT3 activity. Thus, gut microbiota-derived NMN protects against acute pancreatitis [120]. These results suggest that the gut microbiota, the gut microbiota-derived NMN, and the intestinal NAMPT–NAD^+^–SIRT axis collaborate to regulate intestinal homeostasis and systemic metabolic functions (Figure. 3).

HFD intake causes an imbalance in the gut microbial community, affecting bile acid abundance and composition [162]. Several bile acid species, such as tauroursodeoxycholic acid, have disease-preventing effects [163]. Bile acids exert endocrine and metabolic effects by acting on FXR and Takeda G protein-coupled receptor five (TGR5). FXR inhibition increases GLP-1 secretion in response to glucose intake. Thus, bile acid profile alteration affects postprandial GLP-1 secretion and systemic glucose metabolism [164]. In addition, GLP-1 secretion is positively correlated with postprandial bile acid concentration in healthy individuals [165]. Therefore, diet, and its associated gut environment, may have regulatory effects on intestinal homeostasis, GLP-1 secretion, and postprandial glucose homeostasis at least partly by modulating the AMPK–NAMPT–NAD^+^–SIRT1–gut microbiota–Wnt signaling, either dependent or independent of bile acid composition (Figure 3). 

## 6. Therapeutic Potential of NAD^+^ Intermediates as GLP-1 Stimulants

NAD^+^ intermediates, such as NMN and NR, have attracted attention as a strategy to boost NAD^+^ biosynthesis. NMN and NR are endogenously biosynthesized metabolites detected in human breast milk [166]. NMN is also found in edamame, broccoli, cucumber, avocado, tomatoes, beef, and shrimp in small quantities [167]. NMN and NR administration increases NAD^+^ levels in major metabolic organs in animal models of various diseases, exerting remarkable effects on age- and obesity-associated diseases such as diabetes, CVD, cancer, and Alzheimer’s disease [62,168,169,170]. For example, the intraperitoneal administration of NMN restores NAD^+^ biosynthesis and improves glucose intolerance, insulin resistance, and dyslipidemia in age- or HFD-induced diabetic mice by activating SIRT1-catalyzed reactions [42]. Oral administration of NR also increases NAD^+^ levels in the liver, brown adipose tissue, and skeletal muscle. Enhancing NAD^+^ biosynthesis promotes SIRT1- and SIRT3-catalyzed reactions, improving metabolic disorders such as weight gain, insulin resistance, and dyslipidemia associated with an HFD [171]. 

The effects of the NAD^+^ intermediates on longevity and healthy life expectancy have recently gained attention. For example, 6-week NR administration improves muscle stem cell function and prolongs the life span of 24-month-old mice. These findings indicate that boosting NAD^+^ biosynthesis is associated with a life span [172]. Twelve-month oral administration of NMN suppresses weight gain and improves insulin sensitivity, adipose tissue inflammation, physical activity, skeletal muscle mitochondrial function, retinal function, and bone density in 5–17-month-old mice [167]. The details of the in vivo pharmacokinetics of NR and NMN remain unclear. Systemically administered NR may be metabolized in the liver or rapidly hydrolyzed in the blood or intestinal tract and taken up by organs as NAM [43,173,174]. The Slc12a8-encoded NMN-specific transporter has been recently demonstrated to be highly expressed in the small intestine [175], leading to the gradual understanding of the pharmacokinetics of NMN [42,167]. 

A recent study demonstrated that oral administration of NR could affect intestinal NAD^+^ biosynthesis, mitigating ethanol-induced intestinal epithelial barrier damage by protecting mitochondrial function in a SIRT1-dependent manner [176]. We also recently discovered that HFD impairs intestinal NAMPT-mediated NAD^+^ biosynthesis and disrupts the Wnt signaling pathway, GLP-1 production, and whole-body glucose metabolism. HFD-fed obese mice were administered NMN (500 mg/kg/bodyweight/day) using oral gavage for 14 days. NMN significantly increased NAD^+^ concentrations and proglucagon expression in the ileum. In addition, ileum explants from NMN-treated HFD-fed obese mice exhibited higher GLP-1 secretion, which improved postprandial hyperglycemia [31]. Our findings demonstrated that NMN promotes intestinal GLP-1 secretion by restoring intestinal NAD^+^ biosynthesis in HFD-induced obese mice, suggesting that NMN could be a novel therapeutic option for improving postprandial hyperglycemia in obesity. 

Based on these experimental animal data, clinical trials investigating the safety and efficacy of NAD^+^ intermediates in humans are currently underway worldwide. A single oral dose of NR increases the concentration of NAD^+^ metabolites in a concentration-dependent manner in human peripheral blood mononuclear cells (PBMCs) [177]. Furthermore, the safety of its long-term oral administration has been confirmed [178,179]. NR administration benefits blood pressure, carotid-femoral pulse waves, maximal muscle strength, and fatigue in healthy subjects [180]. In elderly and obese subjects, it improves inflammatory blood markers, energy metabolism at bedtime, and fatty liver [181,182,183], though not insulin sensitivity, glucose tolerance, or skeletal muscle mitochondrial function [181,182,184]. In addition, the oral administration of NR does not affect GLP-1 secretion in nondiabetic individuals with obesity [184]. 

The safety and efficacy of NMN administration have been rigorously explored (Table 1). The safety study of a single oral intake of 100, 250, and 500 mg NMN in healthy middle-aged men was conducted in Japan. NMN dose-dependently increased NAM metabolites, such as *N*-methyl-2-pyridone-5-carboxamide and *N*-methyl-4-pyridone-5-carboxamide in the plasma without exerting any significant deleterious effects [185]. In healthy Japanese participants between 20 and 65 years old, oral supplementation with 125 mg NMN twice daily for 12 weeks increased NAD^+^ levels in whole blood without causing abnormalities in physiological and laboratory tests [186]. Oral administration of 1250 mg NMN once daily for four weeks was reported to be safe and well-tolerated in healthy adult men and women aged 20–65 years [187]. In older male patients over 65 years old with diabetes and impaired physical performance, 250 mg NMN supplementation for 24 weeks was safe and tolerated and did not improve grip strength and walking speed [188]. However, an oral intake of 250 mg NMN once daily in elderly men over 65 years old for 12 weeks increased whole blood NAD^+^ and NAD^+^-related metabolite levels, improving gait speed and performance in the left grip test without any deleterious effects [189]. Furthermore, an oral intake of 250 mg NMN once daily in the afternoon for 12 weeks effectively improved physical performance and fatigue in elderly people over 65 years old [190]. Exercise combined with 300, 600, and 1200 mg of daily NMN supplementation dose-dependently increased aerobic capacity in healthy amateur runners [191]. Oral administration of 300, 600, and 900 mg NMN daily for 60 days was well-tolerated and increased blood NAD^+^ concentrations in healthy middle-aged adults, improving physical performance, blood biological age, and subjective general health assessment. However, insulin resistance assessed using the homeostasis model assessment-estimated insulin resistance was not ameliorated [192]. In contrast, oral intake of 300 mg NMN once daily after breakfast for 60 days increased the possibility that NMN could benefit insulin sensitivity in healthy subjects between the ages of 40 and 65 years [193]. Consistent supplementation with 250 mg NMN daily for 10 weeks increased NAD^+^ and its metabolites in PBMCs and improved muscle insulin sensitivity in obese postmenopausal women with prediabetes [194]. 

Based on these findings, the safety of NMN has been demonstrated to a certain extent; however, its effects on whole-body glucose metabolism in humans remain inconclusive. Although oral administration of NR did not improve insulin sensitivity or affect GLP-1 secretion, oral intake of NMN alleviated insulin resistance in obese prediabetic females [194]. As oral NMN administration augments GLP-1 secretion and improves postprandial glucose metabolism in obese mice [31], the different effects between NMN and NR on GLP-1 production and whole-body glucose metabolism may involve the presence of the NMN transporter in the small intestine [175]. Human studies investigating the effects of orally administered NMN on intestinal NAD^+^ biology, GLP-1 secretion, and postprandial glucose metabolism should be comprehensively conducted. The results from the human clinical study will further validate that boosting intestinal NAD^+^ biosynthesis by orally administering NMN could be a new therapeutic approach to improve intestinal homeostasis, postprandial glucose metabolism, and ultimately, healthy life expectancy.

## 7. Conclusions and Prospects

Postprandial hyperglycemia in obesity and aging is an important treatment target to reduce the risk of progression to type 2 diabetes and the incidence of CVDs [1,2,3,4,5,6]. Several medications for postprandial hyperglycemia, such asαGI, glinide, DPP4 inhibitors, and GLP-1 analogs, have been commercially available for the past decade and have been shown to exert efficient antihyperglycemic effects and additional protective effects against CVDs [9,10,24]. However, they can induce adverse effects, and there is a lack of evidence to support their ability to promote healthy aging and life expectancy [11,12,197]. Therefore, the development of novel therapies targeting postprandial hyperglycemia, which could improve both cardiovascular outcomes and healthy life expectancy, has been sought.

The overall findings of several rodent studies targeting intestines have indicated that the intestinal AMPK–NAMPT–NAD–SIRTs axis plays an important role in maintaining intestinal homeostasis, thereby regulating GLP-1 production and postprandial glucose metabolism. Thus, these findings provide important mechanistic and therapeutic insights into the disruption of intestinal homeostasis and postprandial glucose dysregulation, particularly in obesity and aging. Nevertheless, several important questions remain unanswered and require further investigation. First, elucidating the molecular mechanism underlying the regulation of intestinal *Nampt* expression is warranted. Furthermore, given that calorie restriction-induced AMPK activation stimulates *Nampt* transcription [94], and metformin, an AMPK activator, activates intestinal AMPK [93,106,107,108], the effects of metformin administration on *Nampt* expression and GLP-1 production should be comprehensively tested in HFD-fed obese mice with impaired intestinal NAMPT-mediated NAD^+^ biosynthesis [31]. In different cell types, *Nampt* gene transcription is regulated by the circadian locomotor output cycles kaput/brain and muscle ARNT-like one (CLOCK:BMAL1) complex [38,198]. As the ileal diurnal rhythms of microbiome and transcriptome are disrupted in HFD-fed obese mice [199], changes in intestinal *Nampt* transcription and intestinal bacterial composition should be examined in mice with disturbed lifestyles such as diet and the timing of meals. Second, the involvement of gut microbiota and NMN transporter in NMN absorption should be investigated. Orally administered NAM is first converted to nicotinic acid (NA) by the gut microbiota and absorbed as NA from the colon [200], and orally administered NR is converted to NAM and then to NA by the microbiota [201]. These findings suggest that the uptake of orally administered NAD^+^ precursors requires microflora. Consistently, normal gut microbiota variation is required to produce NMN, enhancing NAD^+^ biosynthesis in remote organs [120]. Therefore, the Slc12a8 NMN transporter and microbiota in the small intestine might work synergistically to regulate intestinal and whole-body NAD^+^ biosynthesis. Thus, NMN transporter expression, microbiota composition, and NAD^+^ metabolism in multiple organs in INKO mice should be further explored. Lastly, the effects of NMN on GLP-1 production, postprandial glucose regulation, gut microbiota, and healthy life expectancy in aged individuals with obesity should be evaluated in randomized, controlled clinical trials. Such studies will elucidate the novel therapeutic potential of intestinal NAD^+^ biology for improving postprandial hyperglycemia and healthy longevity.

## Figures and Tables

**Figure 1 nutrients-15-01494-f001:**
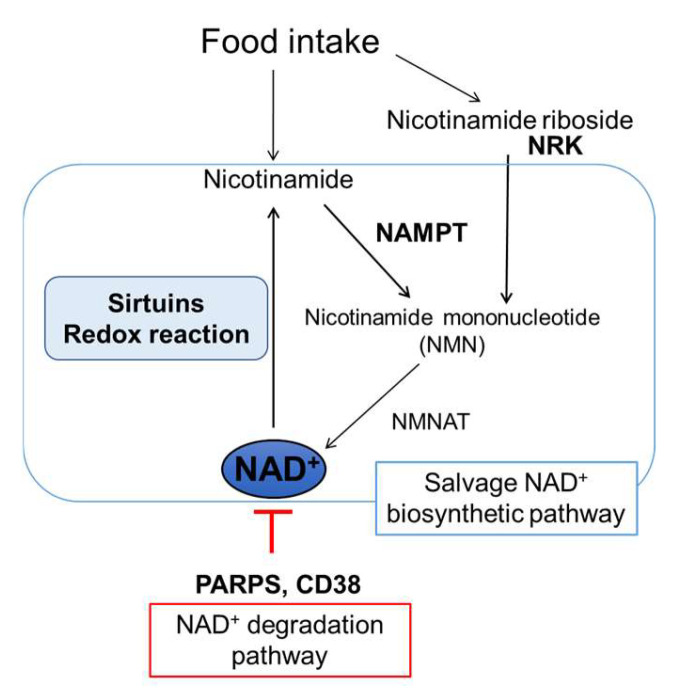
Mammalian NAD^+^ biosynthetic salvage and degradation pathways. In mammals, NAMPT is the key enzyme in NAD^+^ biosynthetic salvage pathway initiated from nicotinamide in food. SIRTs consume NAD^+^ in a tissue-dependent manner, and NAD^+^ degradation is mediated by PARPs and CD38. NMNAT—nicotinamide/nicotinic acid mononucleotide adenylyltransferase; NRK—nicotinamide riboside kinase; NAMPT—nicotinamide phosphoribosyltransferase; SIRTs—sirtuins; PARPs—poly ADP ribose polymerases; CD38—cluster of differentiation 38; NAD^+^—nicotinamide adenine dinucleotide.

**Figure 2 nutrients-15-01494-f002:**
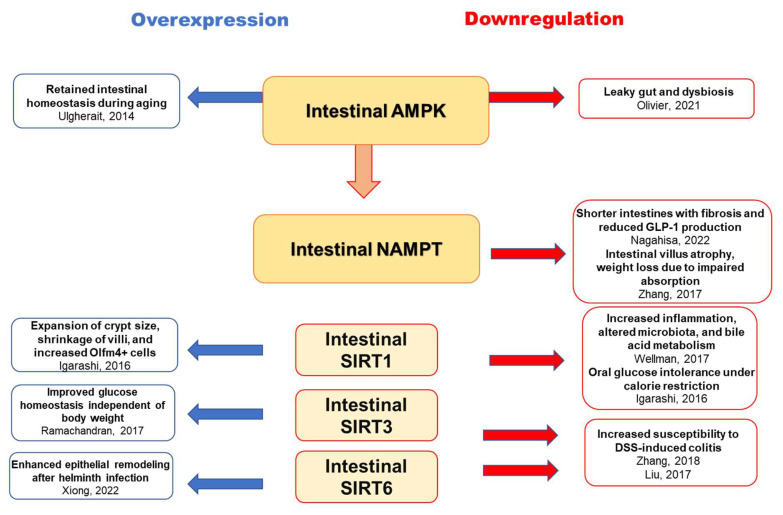
Intestinal phenotype induced by the overexpression or knockout of the intestinal AMPK-NAMPT-NAD^+^-sirtuin pathway [31,87,94,95,98,101,102,103,104,105]. Olfm4—olfactomedin 4.

**Figure 3 nutrients-15-01494-f003:**
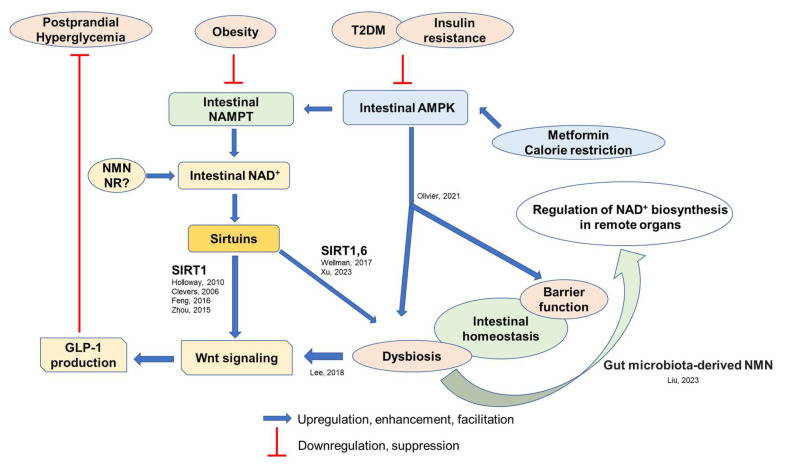
Vicious cycles of metabolic disorders by disrupting intestinal AMPK–NAMPT–NAD–SIRT1–gut microbiota–Wnt signaling–GLP-1 axis [98,101,115,116,117,118,119,120,121]. NMN—nicotinamide mononucleotide; NR—nicotinamide riboside; Wnt—wingless-related integration site; GLP-1—glucagon-like peptide-1.

**Table 1 nutrients-15-01494-t001:** Summary of recent NMN human clinical studies.

Design	Subjects	Dose, Administration Route, and Form of NMN	Treatment Duration	Method for Validating Increased NAD^+^ Level	Findings	Publication Year	Ref.
Single-arm, nonrandomized, nonblinded study	Age, 40–60 years; healthy men (*n* = 10)	100, 250, and 500 mg/day; oral administration; capsule	Single administration	NMN administration dose-dependently increased plasma concentrations of 2PY and 4PY.	Single oral administration of NMN did not cause significant changes in clinical symptoms, including vital signs, ophthalmic examination, sleep quality, and laboratory analysis results.	2020	[185]
Randomized, double-blind, placebo-controlled, parallel-group study	Age, 20–65 years; healthy males and females (NMN, *n* = 15; placebo, *n* = 15)	250 mg/day; oral administration; tablet	12 weeks	NMN administration increased NAD^+^ and NAMN levels in whole blood.	No obvious abnormalities in physiological and laboratory tests and no adverse effects were observed.	2022	[186]
Randomized, double-blind, placebo-controlled, parallel-group study	Age, 20–65 years; healthy males and females (NMN, *n* = 16; placebo, *n* = 15)	1250 mg /day; oral administration; packaged powder dissolved in water (200 mL)	4 weeks	N/A	Oral administration of NMN 1250 mg/day for 4 weeks did not cause significant abnormalities in anthropometry, hematological, biochemical, urine, and body composition analyses.	2022	[187]
Prospective, placebo-controlled, double-blind study	Age, over 65 years; elderly males with type 2 diabetes with reduced grip strength or walking speed (NMN, *n* = 6; placebo, *n* = 7)	250 mg/day; oral administration; capsule	24 weeks	N/A	Adverse events were not observed in the NMN group. NMN did not improve grip strength and walking speed. However, an improved prevalence of frailty and central retinal thickness was observed.	2023	[188]
Randomized, double-blind, placebo-controlled, parallel-group study	Age; over 65 years; elderly healthy males (NMN, *n* = 11; placebo, *n* = 11)	250 mg/day; oral administration; pill	12 weeks	NMN administration increased NAD^+^ and NAD^+^-related metabolites levels in whole blood, assessed by metabolomic analysis	NMN nominally but significantly improved gait speed and performance in the left grip tests without affecting body composition and glucose metabolism were observed.	2022	[189]
Randomized, double-blind, placebo-controlled study	Age, over 65 years; elderly males (NMN antemeridian, *n* = 27; post meridian, *n* = 27. Placebo antemeridian, *n* = 27; post meridian *n* = 27)	250 mg/day; oral administration; tablet	12 weeks	N/A	NMN intake in postmeridian improved lower limb function and drowsiness.	2022	[190]
Randomized, double-blind, placebo-controlled, four-arm clinical study	Age, 27–50 years; healthy recreationally trained runners (40 males and 8 females)	300, 600, 1200 mg/day, oral administration; powder	6 weeks	N/A	Exercise combined with 300, 600, and 1200 mg of daily NMN supplementation dose-dependently increased aerobic capacity.	2021	[191]
Randomized, double-blind, placebo-controlled, parallel-group study. Dose-dependent study	Age, 40–65 years; healthy males and females (NMN, 300, 600, and 900 mg; placebo, *n* = 20)	300, 600, 900 mg/day; oral administration; capsule	60 days	Blood NAD^+^ concentrations were increased compared to baseline in three NMN-treated groups (300, 600, 900 mg) on days 30 and 60.	NMN administration was safely tolerated. Walking distance during the six-minute walking test, the change of biological age, and SF-36 scores were improved in the NMN 300, 600, and 900 mg groups on day 60.	2023	[192]
Randomized, double-blind, placebo-controlled, parallel-group study	Age, 40–65 years; healthy males and females (NMN, *n* = 31; placebo, *n* = 35)	300 mg/day; oral administration; capsule	60 days	Serum NAD^+^/NADH levels were increased by 11.3% on day 30 and 38% on day 60 vs. baseline.	Walking endurance: SF-36 questionnaire score, a parameter for well-being, and HOMA-IR index improved with NMN administration for 60 days.	2022	[193]
Randomized, double-blind, placebo-controlled study	Age, 55–75 years; postmenopausal women with prediabetes (NMN, *n* = 13; placebo, *n* = 12)	250 mg/day; oral administration; capsule	10 weeks	Plasma concentrations of 2 PY and 4 PY and NAD^+^ contents in PBMCs increased after 10 weeks of NMN treatment. *N*-methyl-nicotinamide, 2PY, and 4PY increased in quadriceps muscle tissue samples obtained 1.5 h after the last dose of NMN	NMN increased muscle insulin sensitivity assessed using the hyperinsulinemic-euglycemic clamp.	2021	[194]
Single-blind study	Age, 45–75 years; males and females with sleep disturbance without primary conditions (NMN, *n* = 32; placebo, *n* = 31)	300 mg/day; oral administration; capsule	12 weeks	N/A	NMN improved sleep quality assessed using PSQI and smart bands sleep data	2022	[195]
Open-label, single-arm exploratory study	Age, 20–70 years; healthy males and females (*n* = 10)	300 mg/day; intravenous administration; dissolved in saline (100 mL)	Single administration	Total amount of NAD^+^ levels in the blood was increased.	Intravenous NMN administration reduced blood triglyceride levels without affecting blood cells, electrocardiograms, pulse, blood pressure, and metabolic markers in the liver, heart, pancreas, and kidneys.	2022	[196]

2PY—*N*-methyl-2-pyridone-5-carboxamide; 4PY—*N*-methyl-4-pyridone-5-carboxamide; PBMC—peripheral blood mononuclear cell; N/A—not applicable; NAMN—nicotinic acid mononucleotide; HOMA-IR—homeostatic model assessment for insulin resistance; SF-36—36-item short form survey instrument; PSQI—Pittsburgh sleep quality index.

## Data Availability

Not applicable.

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
