# Peer review of "Interactions between Intestinal Homeostasis and NAD+ Biology in Regulating Incretin Production and Postprandial Glucose Metabolism"

_nutrients, 2023, doi:10.3390/nu15061494_

Round 1

Reviewer 1 Report

The review by Nagahisa et al. on the topic of the intestinal AMPK-NAMPT-NAD+-SIRT pathway in relation to intestinal homeostasis, GLP-1 secretion, and postprandial glucose metabolism is a very detailed, well-structured, and well-written review. While the review is very easy to understand, the figures are very "basic". Here I would suggest to modify especially figure 2. For example, the role of the intestinal AMPK/NAMPT/NAD/SIRT signaling pathway should be more clearly elaborated. Currently, the whole pathyway is summarized in one box, although it is the main topic of the review. Also, the effect of the AMPK/NAMPT/NAD/SIRT pathway on intestinal homeostasis should be more clearly elaborated in this figure, currently only an arrow leads to the microbiota, here it is not clear how this is influenced.
Further minor suggestions:
Another suggestion would be to include a figure on NAD+ biosynthesis (and possibly degradation), the process is described in the text in section 2.1 but would certainly be more understandable with a figure.
Table 1: Please also include the publication date to better classify the term "recent" in the title of the table. It would also be interesting to know in which form of oral administration NMN was used. As a pill? As a capsule? In food?

Author Response

Point-by-point responses to reviewers’ comments and suggestions are provided below:

 Manuscript ID: nutrients-2265805

 Interactions between intestinal homeostasis and NAD+ biology in regulating incretin production and postprandial glucose metabolism

We thank the editor and reviewers for their insightful and constructive review, which has helped to significantly improve the manuscript. The revised manuscript includes changes made in response to the comments, which are marked in red font for your convenience.

The review by Nagahisa et al. on the topic of the intestinal AMPK-NAMPT-NAD+-SIRT pathway in relation to intestinal homeostasis, GLP-1 secretion, and postprandial glucose metabolism is a very detailed, well-structured, and well-written review.

Reply: We express our gratitude for the positive comments and acknowledge your description of our review article as “very detailed, well-structured, and well-written”.

While the review is very easy to understand, the figures are very "basic". Here I would suggest to modify especially figure 2. For example, the role of the intestinal AMPK/NAMPT/NAD/SIRT signaling pathway should be more clearly elaborated. Currently, the whole pathyway is summarized in one box, although it is the main topic of the review. Also, the effect of the AMPK/NAMPT/NAD/SIRT pathway on intestinal homeostasis should be more clearly elaborated in this figure, currently only an arrow leads to the microbiota, here it is not clear how this is influenced.

Reply: Thank you for your valuable suggestion. We agree with your viewpoint on this. As a result, modifications have been made to the previous Figure 2, which has been replaced with a new Figure 3 that elaborates on how AMPK/NAMPT/NAD/SIRT interact with each other and regulate intestinal homeostasis.

Further minor suggestions:
Another suggestion would be to include a figure on NAD+ biosynthesis (and possibly degradation), the process is described in the text in section 2.1 but would certainly be more understandable with a figure.

Reply: Thank you for your observation and excellent suggestion. A new Figure 1 has been added, which describes the mammalian NAD+ biosynthetic salvage pathway and its degradation pathway, which is mediated by PARPs and CD38.

Table 1: Please also include the publication date to better classify the term "recent" in the title of the table. It would also be interesting to know in which form of oral administration NMN was used. As a pill? As a capsule? In food?

Reply: Apologies for the omission. The publication year of the articles and the form of NMN administered to the participants in the NMN human clinical studies have been added accordingly. (Table 1)

We thank you for your insightful comments and observations, which have helped strengthen the manuscript. Necessary changes have been made to incorporate your feedback and suggestions in the revised manuscript. We look forward to working with you to bring the manuscript closer to publication in Nutrients.

Sincerely,

Shintaro Yamaguchi, M.D., Ph.D.

Division of Endocrinology, Metabolism and Nephrology, Department of Internal Medicine, Keio University School of Medicine, 35 Shinanomachi, Shinjuku-ku, Tokyo, Japan

Tel: +81-3-5363-3796

Fax: +81-3-3359-2745

E-mail: yama1005@a6.keio.jp.

Reviewer 2 Report

I read with great interest this review, which is extremely well structured and clear. It summarizes the NAMPT-related mechanisms to regulate intestinal homeostasis and GLP-1 secretion. Also, possible therapeutic applications targeting NAD+ biology in this context have been highlighted and discussed.

Since relevant to the review, I only would ask to add reference to a study revealing that CD38 downregulation, with consequent NAD increase in BAT, is important for BAT activity (ref PMID 33010451, can be inserted in line 140 or line 168). Also, beside SIRT1, also SIRT6 had been described to regulate the acetylation status of NAMPT, regulating iNAMPT activity and  eNAMPT release (although in different cells; ref PMID 30514106). This can be added at line 175.

Author Response

Point-by-point responses to reviewers’ comments and suggestions are provided below:

 Manuscript ID: nutrients-2265805

 Interactions between intestinal homeostasis and NAD+ biology in regulating incretin production and postprandial glucose metabolism

 We thank the editor and reviewers for their insightful and constructive review, which has helped to significantly improve the manuscript. The revised manuscript includes changes made in response to the comments, which are marked in red font for your convenience.

I read with great interest this review, which is extremely well structured and clear. It summarizes the NAMPT-related mechanisms to regulate intestinal homeostasis and GLP-1 secretion. Also, possible therapeutic applications targeting NAD+ biology in this context have been highlighted and discussed.

Reply: We express our gratitude for the positive comments provided and also acknowledge your description of our review article as “well structured and clear”.

Since relevant to the review, I only would ask to add reference to a study revealing that CD38 downregulation, with consequent NAD increase in BAT, is important for BAT activity (ref PMID 33010451, can be inserted in line 140 or line 168).

Also, beside SIRT1, also SIRT6 had been described to regulate the acetylation status of NAMPT, regulating iNAMPT activity and eNAMPT release (although in different cells; ref PMID 30514106). This can be added at line 175.

Reply: Thank you for your careful observation and valuable suggestion. Reference in-text citations to these articles have been added in the revised manuscript (Lines 150–152 and 188–192, respectively).

 We thank you for your insightful comments and observations, which have helped strengthen the manuscript. Necessary changes have been made to incorporate your feedback and suggestions in the revised manuscript. We look forward to working with you to bring the manuscript closer to publication in Nutrients.

Sincerely,

Shintaro Yamaguchi, M.D., Ph.D.

Division of Endocrinology, Metabolism and Nephrology, Department of Internal Medicine, Keio University School of Medicine, 35 Shinanomachi, Shinjuku-ku, Tokyo, Japan

Tel: +81-3-5363-3796

Fax: +81-3-3359-2745

E-mail: yama1005@a6.keio.jp.
